# Rare Earth Cerium Increases the Corrosion Resistance of NdFeB Magnets

**DOI:** 10.3390/ma13194360

**Published:** 2020-09-30

**Authors:** Jialei Dai, Zixuan Yang, Qian Liu

**Affiliations:** 1Key Laboratory of Impact and Safety Engineering, Ministry of Education, Ningbo University, Ningbo 315211, China; zjdaijialei@163.com; 2Key Laboratory of the Ministry of Education for Modern Metallurgy Technology, North China University of Science and Technology, Tangshan 063210, China; liuqian@ncst.edu.cn

**Keywords:** NdFeB magnets, corrosion, SKPFM, EIS, rare earth Ce

## Abstract

Herein, we investigated the effects of Ce on the corrosion behavior of NdFeB magnets in 3.5% NaCl solutions using electrochemical tests, scanning electron microscopy (SEM), energy dispersive spectroscopy (EDS) mapping, and scanning Kelvin probe force microscopy (SKPFM). We demonstrated that Ce markedly enhances the corrosion resistance of NdFeB magnets. Ce primarily replaces Nd in the Nd-rich phase instead of matrix phase, increasing the surface potential of the Nd-rich phase. An increase in the Ce content from 0 to 5.21 wt%, decreased the potential difference between the main phase and (Nd, Ce)-rich phase from 350.2 mV to 97.7 mV; therefore, the corrosion resistance of the magnetic materials increased. The corrosion resistance constituted the Nd-rich phase < the void < metal matrix. Moreover, based on the results of the study, we discussed the impact mechanism of additions of Ce on the corrosion resistance of the magnets.

## 1. Introduction

Sintered NdFeB magnets have been widely used in the manufacturing of gadgets and equipment in various fields such as information technology, communication electronics, and wind power generation, among other fields. This is because of its ultra-high permanent magnetic properties. However, its scope of application is limited by its poor corrosion resistance. Considering this, improving its corrosion resistance without affecting its magnetic properties has been the main research focus of many scholars [1,2,3].

Sintered NdFeB magnets are composed of three phases: the main phase (Nd_2_Fe_14_B), Nd-rich phase, and the B-rich phase (Nd_1.1_Fe_4_B_4_). The oxidation capacity of NdFeB permanent magnetic material is often attributed to the strong oxidation ability of the rich B and rich Nd phase. This is because the three phases have different oxidation capacities. The B-rich and Nd-rich phases are generally distributed on the grain boundary. As such, the oxidation of the grain boundary results in its corrosion. Moreover, the porous nature of sintered NdFeB magnets affects their ability to form oxide protective film on their surfaces. This causes them to corrode [4,5].

Meakin et al. reported that corrosion resistance, magnetic properties, and temperature stability of NdFeB magnets vary greatly after alloy elements are added to them [6]. Alloying elements are usually added as substituted or doped elements. A substitution element is added to the magnet to replace some atoms in the Nd_2_Fe_14_B phase to improve the intrinsic properties of the main phase. However, the soft magnetic contact generated reduces the magnetic energy product and remanence of the magnet. Doping elements cause exsolution within the hard-magnetic square phase and the grain boundary. They can also cause exsolution with in the main phase to form a new phase that replaces the original rich Nd or rich B phase. These changes can achieve the goal of improving the hard-magnetic grain boundaries’ microstructure and magnetic and corrosion-resistant properties [7]. Doping alloy elements added to NdFeB are divided into two categories: the first category comprises aluminum, copper, zinc, gallium, germanium, and tin, while the second comprises vandium, molybdenum, tungsten, titanium, and zirconium. The first category formd Nd-Fe-m intercrystalline phase while the second category form m-B or Fe-m-B intercrystalline phase [8]. The intermetallic compounds formed at the grain boundary replace the original B-rich phase, thus preventing the grain growth of the tetragonal phase. In addition, these new phases partially replace the Nd-rich phase, thereby reducing the potential differences between the phases of the magnet. This in turn improves the corrosion resistance of the metal boron magnet [9,10,11].

Elaziz et al. studied the influence of Al, Co, Mo, and other elements on the corrosion performance of magnets [12]. They found that V formed the (V_1−X_Fe_x_)_3_B precipitation phase with Fe-V precipitation particles at the grain boundary, Mo formed the (Mo_1−x_FeX)_3_B precipitation phase, while Co formed the Nd_3_Co intermediate phase in the Nd-rich grain boundary phase. These phases partially replaced the Nd-rich phase at the grain boundary, thereby improving the corrosion resistance of the magnet. Al and Co replaced Fe in the Nd_2_Fe_14_B phase. The substitution of Fe with either of the two elements greatly inhibited the oxidation and corrosion behavior of the main phase at temperatures above 200 °C. Knoch et al. found that NdFeB magnets formed relatively stable intermetallic compounds at the grain boundary after adding defined aluminum and gallium quantities to NdFeB magnets. The stable intermetallic compounds were smooth and surrounded by the main phase. These large and stable intermetallic compounds greatly improved the corrosion and high-temperature resistance of the magnets [13]. In the same line, Yu et al. found that the addition of dysprosium and niobium formed a more stable intercrystal line phase near the grain boundary of the magnet, thereby improving the electrochemical potential of the intercrystal line phase. Moreover, the addition of the elements refined the grains of the magnet thus greatly improving the corrosion resistance of the magnet. Furthermore, the increase in Nb quantity led to a significant reduction in the grain size of the main phase, thereby causing the grain boundary of the NdFeB magnet to be clear, smooth, and evenly distributed around the main phase. This in turn improved the corrosion resistance of the magnet [14].

The rapid growth of the scope of applications of sintered NdFeB magnets increase the demand and price of the rare-earth Nd. Among the rare earth elements, cerium is more readily available than the rest. As such, the utilization of Ce in the rare earth permanent magnet industry can be of huge economic benefit [15,16]. Thus, it is of great significance to develop low-cost sintered NdFeB magnets and study their corrosion resistance.

However, the corrosion behavior of Ce-doped NdFeB magnets remains unknown. Herein, the effects of Ce on the corrosion behavior of Ce-doped NdFeB magnets in 3.5% NaCl solutions was investigated by electrochemical tests, scanning electron microscopy (SEM), energy dispersive spectroscopy (EDS) analysis, and scanning Kelvin probe force microscopy (SKPFM).

## 2. Experiment

### 2.1. Materials and Sample Preparation

Didymium, praseodymium, iron, ferroboron, and dysprosium with a purity of 99.5% were utilized as raw materials and melted in a vacuum induction furnace to form alloy ingots. After that, the ingot was peeled and coarsely broken, then the powder was ground under the protection of N_2_. The NdFeB magnetic material was pressed into shape after the direction of the magnetic field was determined.

After sintering at 1050 °C for 200 min and tempering at 630 °C for 150 min, the raw blanks were put into a high vacuum sintering furnace and prepared into sintered NdFeB magnets with Ce contents of 5.02, 3.11, and 0 wt%. The Chemical compositions of (Nd, Ce)FeB are listed in Table 1.

We cut samples (ϕ10 mm size) for electrochemical analysis from the NdFeB magnets agglomerates. We used silica gel to seal the non-working surface of the samples. In the weight loss analysis, the corrosion samples with 40 mm × 20 mm × 3 mm dimensions were used. We utilized an emery paper to sand the working surface of the samples from 400 to 5000 in number and then polished them using a 1 μm diamond paste. The effects of surface roughness on pitting corrosion were avoided. Finally, we cleaned the samples ultrasonically in ethanol. All the tests were performed at room temperature and at a pH of 7 [17].

### 2.2. Electrochemical Measurements

We measured the electrochemical measurements in 3.5% NaCl solution using the Gamry electrochemical workstation (Gamry Instruments, Warminster, PA, USA). Briefly, we prepared the solution using analytical grade chemicals and deionized water. The electrochemical test device used in this study consisted of a three-electrode system. Platinum foil and Ag/AgCl electrode constituted the counter electrode and the reference electrode, respectively. We performed the in-situ electrochemical impedance spectroscopy (EIS) experiments during the immersion test at a frequency of between 0.01 Hz and 100 kHz with a 5-mV amplitude signal at open circuit potential. We used the ZSimpWin software (2010, AMETEK, San Diego, CA, USA) to analyze the experimental data. We measured the potentiodynamic polarization curves at room temperature, and the scanning rate was 0.5 mV/s. Moreover, we tested the open circuit potential (OCP) for 30 min [18].

### 2.3. Microscopy Observations

We used SEM (Zeiss, EVO MA 10/LS 10, Oberkochen, Germany) equipped with the energy dispersive spectroscopy (EDS) to observe the microstructure. The morphology and composition were analyzed using SEM and EDS mapping, respectively. We also utilized atomic force microscopy (AFM, Agilent, 5500 AFM/SPM, Santa Clara, CA, USA) to analyze the surface morphology as well as the contact potential difference (CPD) around the Nd-rich phase in the magnet. After that, we performed scanning Kelvin probe force microscopy (SKPFM, Agilent, 5500 AFM/SPM, Santa Clara, CA, USA) on the polished samples. The specimens were marked using a micro-hardness tester before the SKPFM tests. We then used the line scan analysis of 512 × 512 images acquired at 1 Hz to obtain the contact potential around inclusions. These magnets were investigated via X-ray diffraction (XRD) with Cu-Ka radiation [19].

## 3. Results

The Open circuit potential (OCP) of the NdFeB with different Ce contents submerged in 3.5% NaCl solution for 1800 s is shown in Figure 1. At the early submersion stage, the potential declined abruptly. Extending the submersion time reduced the potential slightly and then stabilized gradually. Lastly, the OCP of the magnets with a higher Ce content had a somewhat elevated value compared with the magnets with a lower Ce content [20,21,22,23].

Previous works have found that the higher open circuit equilibrium potential, the higher the corrosion resistance of the metal. Therefore, we can conclude that rare earth Ce can improve the corrosion resistance of magnetic materials in NaCl solution [24,25].

The potentiodynamic polarization curves of the tested magnets in 3.5% NaCl solutions are shown in Figure 2. Our results revealed that the starting point of the anode side of the polarization curves is modulated via active dissolution reactions, and the anodic current density intensified rapidly with the augmented anode potential. The primary data isolated from the curves are shown in Table 2. Our findings indicate that the current density I_corr_ of the corrosion declined sharply from about 3.6 × 10^−8^ A/cm^2^ for Ce = 0 wt% to approximately 1.9 × 10^−8^ A/cm^2^ for Ce = 6 wt%. The potential E_corr_ of corrosion shifted towards the positive direction at about 68.9 mV and 153.2 mV after adding 3% and 5% Ce, respectively. This could be attributed to the effect of the enhancement on the anodic reaction. The anodic Tafel curve, βa, gradually increases with Ce content, indicating that Ce promotes the anodic reaction. At more positive potential than the potential of the corrosion E_corr_, the addition of metal Ce leads the anode polarization curve to a smaller current density. The current density of magnets without Ce appear to be smaller which can refer to smaller corrosion rate. This implies that the additions of Ce inhibit the anodic corrosion processes [26,27].

The EIS results of the specimens of (Nd, Ce)FeB magnets with varied levels of Ce in 3.5% NaCl solution are shown in Figure 3. The Nyquist findings indicated that the decrease in the composition of Ce remarkably increases the diameter of the capacitive loop. In the Nyquist diagram, one capacitive loop from high to medium frequencies, as well as one inductance loop at low frequencies are shown. Generally, the capacitive loops are associated with the reaction in the corrosion product films as well as the electric double layer between the electrode and solution interface [28]. Besides, the larger the diameter of the semicircle, the higher the resistance to corrosion. At the same time, we observed low-frequency inductive loops in all the specimens. The presence of the inductive loop showed the manifestation of adsorption and desorption for the transition action in electrochemical corrosion. In a NaCl aqueous solution, the low-frequency inductive loop of NdFeB could have emanated from the escalated pitting corrosion on the electrode surface because of Cl adsorption [18]. The presence of inductance implies that Cl causes a regular and stable electrochemical reactive channel in fixed positions on the surface of the NdFeB magnet.

As the Ce powders was added to magnets, the stability improvement of intergranular regions will increase the resistance against the formation of active reaction channels while inhibiting the absorption processes of (MeOH) ads. Thus, the corrosion resistance containing Ce magnets is higher than without Ce magnets.

## 4. Discussion

In 3.5 wt% NaCl solution, M^2+^ metal ions were formed in the solution’s electrode system and the grain boundary region of the magnet. Cl^−^ ion in the solution was the readily available anions that M^2+^ could bind to. As such, they concentrated on the surface of the grain boundary phase region and bound the M^2+^ ions. The reaction mechanism is shown in Equation (1):(1)$$M^{2+}+{\mathrm{mH}}_{2}O+{\mathrm{nCl}}^{-}\to {\left[M{\left(\mathrm{OH}\right)}_{m}{\left(\mathrm{Cl}\right)}_{n}\right]}^{2-m-n}+{\mathrm{mH}}^{+}$$

Based on the reaction mechanism shown in Equation (1), the quantity of [M(OH)_m_(Cl)_n_]^2−m−n^ and H^+^ on the electrode surface increased with the progress of the reaction in solution. The number of hydrogen ions remained relatively low during the initial rapid anode polarization stage. Similarly, the quantity of [M(OH)_m_(Cl)_n_]^2−m−n^ was very low on the interface between the grain boundary phase of the magnet and NaCI solution. The interface adsorption phenomenon conformed to the Langmuir isothermal adsorption. The anode current density is shown in Equation (2):(2)$$i_{a}=\mathrm{ka}{\left(\alpha_{H+}\right)}_{x=0}^{-1}\exp\left(1+\beta \right)F\varphi_{a}/\mathrm{RT}$$
where k_a_ is the velocity constant, *β* is the migration coefficient, φ_a_ is the anode potential, α_H+_ is the concentration of H^+^ on the electrode surface of the grain boundary phase, F is the Faraday constant, R is the universal gas constant, and T is the temperature.

In the steady-state anode polarization phase, the number of generated ions increased and continuously diffused into the solution at a speed proportional to the current. Based on the diffusion theory, the current density *i_a_* of the anode is shown in Equation (3):(3)$$i_{a-}=\sqrt{2Fk_{a}D_{H+}/m\delta }\exp\left(1+\beta \right)F\varphi_{a}/2\mathrm{RT}$$
where D_H+_ is the diffusion coefficient of the hydrogen ions and *β* is the thickness of the diffusion layer. 

H^+^ was produced as a result of Cl^−^ ions adsorption in the 3.5 wt% NaCl solution. The corrosion resistance of sintered NdFeB magnets in acidic solution is poor. Thus, the electrochemical corrosion process of sintered NdFeB magnets is closely related to the number of H^+^ formed on the surface of the grain boundary region in the NaCl solution.

EDS mapping results revealed that Nd was partly replaced by Ce in the Nd-rich phase (Figure 4). Ce_2_Fe_14_B was formed by replacing Nd in the Nd-rich phase position and Ce_2_Fe_14_B could improve the coerced force of the magnet. The grain boundary of the magnet remained thick with a high concentration of Nd the Nd-rich phase when no Ce was added to the magnet. The Nd-rich phase easily reacted with H_2_O and Cl^−^ thus generating more H^+^ that accelerated the corrosion of the magnet. When the Ce content was increased to 5%, the grain boundary became thinner and there was an obvious bright white grain boundary to isolate the main phase grain. The Nd-rich phase at the grain boundary also became less distributed. This caused the formation of only few activation reaction channels in the grain boundary region resulting in the low generation of H^+^ in the NaCl solution. Due to this, there was minimal corrosion of the magnet. Evidently, increasing Ce content inhibited the corrosion of the NdFeB magnets.

Sintered NdFeB magnets are composed of three phases: the main phase (Nd_2_Fe_14_B), the Nd-rich phase, and the B-rich phase (Nd_1.1_Fe_4_B_4_). The main phase (Nd_2_Fe_14_B) is the ferromagnetic phase. The Nd-rich phase is distributed at the grain boundary. A small number of B-rich particles are distributed on the surface of the magnet. The potential differences between the three phases are the driving force of corrosion. As such, when the magnet is in an electrochemical environment, the Nd-rich and B-rich phases form local corrosions because of their negative corrosion potential. Moreover, the local corrosion battery is formed by the large cathode and small anode because of the small size of the anode phase (Neodymium-rich and Boron-rich) and the large size of the cathode phase (Nd_2_Fe_14_B). The Nd-rich phase and B-rich phase are located at the grain boundary of the sintered NdFeB magnets which further accelerates corrosion of the grain boundary. The corrosion process is shown in Figure 5.

The electrochemical corrosion rate of sintered NdFeB magnets depends on the kinetic velocity of the related anode and cathode processes. The corrosion rate is determined by Equation (4) [29]:(4)$$S=ki_{x}=k\frac{E_{\alpha }-E_{g}}{P_{\alpha }+P_{g}}$$
where *E**_α_* is the cathode process equilibrium potential, *E_g_* is the anode process equilibrium potential, *P**_α_* is the cathode polarizability, *P_g_* is the anode polarizability, and (*E**_α_*
*− E_g_*) is the electromotive force that corrodes the battery. It is the driving force of the corrosion process. The force is proportional to the free energy reduced in the system during the corrosion process. (*P**_α_* + *P_g_*) is the total kinetic resistance in the electrochemical process.

Based on Equation (4), corrosion speed can be reduced by three sequential steps. The degree of thermodynamic instability is first reduced by bringing the anode process equilibrium potential closer to the cathode process equilibrium potential. This is then followed by hindering of the cathode and anode processes by increasing the polarization of the cathode and anode respectively.

Based on Equation (4), it is evident that the surface potential of a metal is an important parameter in evaluating its corrosion resistance. The activity of the Nd-rich phase in sintered NdFeB magnets expresses this potential. Based on this fact, SKPFM measurements of the three kinds of NdFeB magnets were taken to map Volta potential variations of their polished sample surfaces to determine their corrosion tendency (relative nobility). The morphology and Volta potential images of the three kinds of NdFeB magnets are shown in Figure 6. Volta potential is the potential relative to the test tip. The higher the Volta potential, the greater the difference between the metal and the tip potential, and the lower the corrosion resistance [30,31,32]. The volta potentials of the Nd-rich phases of the three magnetic materials were much higher than the surface potential of their metal matrix. On the other hand, the void was only slightly higher than the surface potential of the metal matrix. The corrosion resistance was the Nd-rich phase < the void < metal matrix. Therefore, the Nd-rich phase and the metal matrix formed a stronger galvanic cell in the solution, the metal matrix as the anode, and the Nd-rich phase as the cathode, thus inducing corrosion.

Comparing the potential diagram of the three kinds of the NdFeB magnets, we established that in the absence of the heavy rare earth Ce, the maximum potential difference between the (Nd, Ce)-rich phase and the main phase was 350.2 mV. Regarding the heavy rare earth contents Ce = 3 wt% and Ce = 5 wt%, the maximum potential differences between the phase and the matrix rich in neodymium were 138.4 mV and 97.7 mV, respectively. The potential difference decreased with the increase of Ce composition. Then, the galvanic cells formed by the (Nd, Ce)-rich phase and metal matrix decreased with the increase in the Ce composition. Therefore, the addition of the heavy rare earth Ce increased the local resistance to corrosion in the NdFeB magnets. These findings are consistent with the results of the impedance experiment.

Ce mainly replaced Nd in the Nd-rich phase instead of Nd in the main phase. The addition of Ce led to the decrease in the surface potential of the Nd-rich phase, which decreased the surface potential difference, hence improving the corrosion resistance of the magnetic materials. The addition of Ce results in a less Nd-rich phase and lower potential difference between the matrix and the Nd-rich phase, thus improving its corrosion resistance.

## 5. Conclusions

We investigated the corrosion behavior of the (Nd, Ce)FeB magnet with varied Ce compositions in 3.5% NaCl solution using electrochemical and SKPFM tests. Our findings revealed that:

(1) The addition of rare earth Ce remarkably improves the resistance to corrosion of the NdFeB magnets. Ce mainly replaces Nd in the Nd-rich phase instead of Nd in the main phase. Moreover, the addition of Ce results in a less Nd-rich phase and lower potential difference between the main phase and the (Nd, Ce)-rich phase. These factors reduce the accelerating force of corrosion and suppress the formation of the active reaction channels in the grain boundaries, resulting in the elevated corrosion resistance of the Ce-doped magnets compared with the Ce-free magnet.

(2) The corrosion resistance was the Nd-rich phase < the void < metal matrix. In the absence of the Ce, the maximum potential difference between the matrix phase and Nd-rich phase was 350.2 mV. For Ce = 3 wt% and Ce = 5 wt%, the maximum potential differences were 138.4 mV and 97.7 mV, respectively.

## Figures and Tables

**Figure 1 materials-13-04360-f001:**
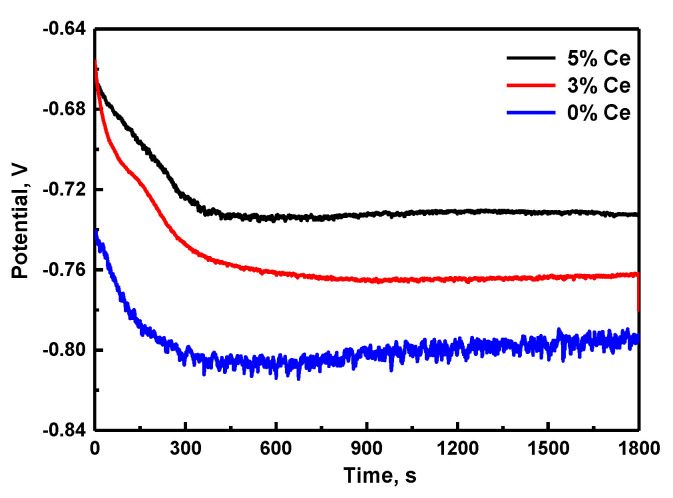
Open circuit potential (OCP) of (Nd,Ce)FeB magnets with different Ce contents in 3.5% NaCl solution for 1800 s.

**Figure 2 materials-13-04360-f002:**
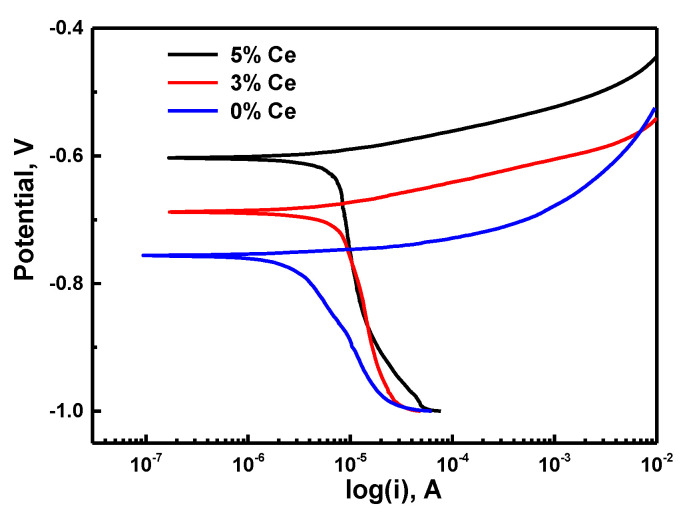
Polarization curves of the tested magnets in 3.5% NaCl solution.

**Figure 3 materials-13-04360-f003:**
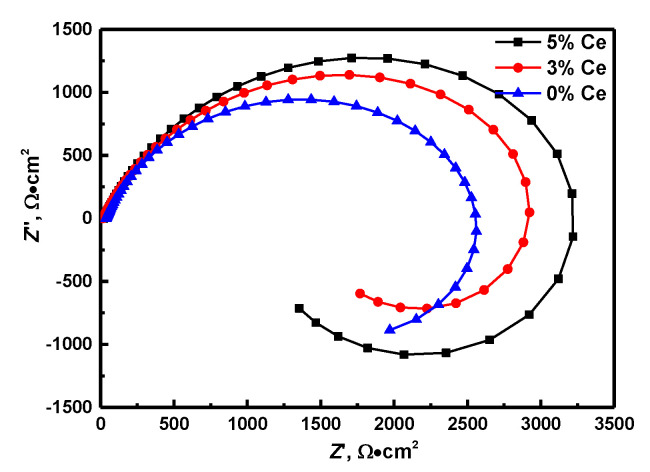
The Nyquist diagrams of NdFeB magnet with different Ce contents in 3.5% NaCl solution.

**Figure 4 materials-13-04360-f004:**
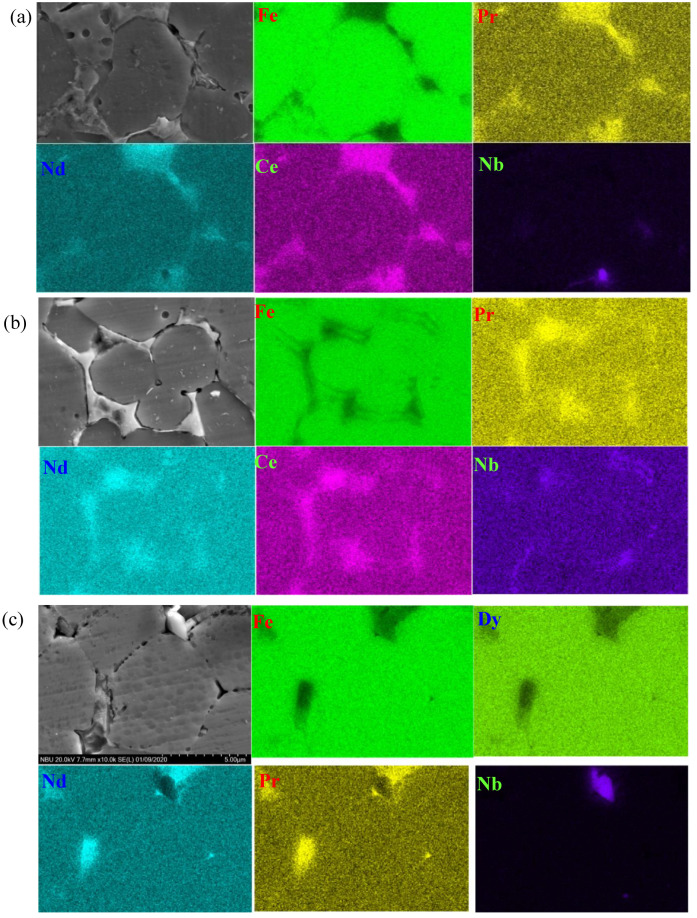
Energy dispersive spectroscopy (EDS) mapping around Nd-rich phase of NdFeB magnet, (**a**) 5% Ce, (**b**) 3% Ce, (**c**) without Ce.

**Figure 5 materials-13-04360-f005:**
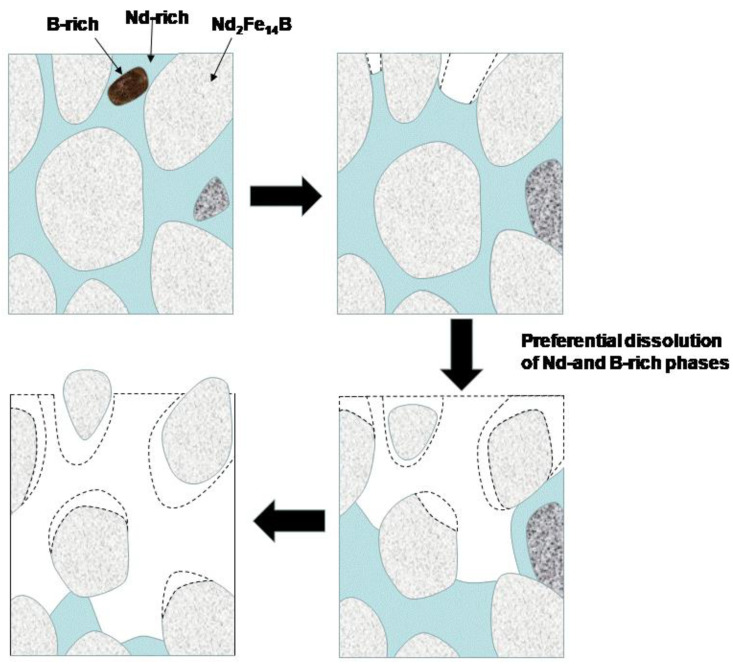
Schematic diagram of sintered NdFeB magnet corrosion process.

**Figure 6 materials-13-04360-f006:**
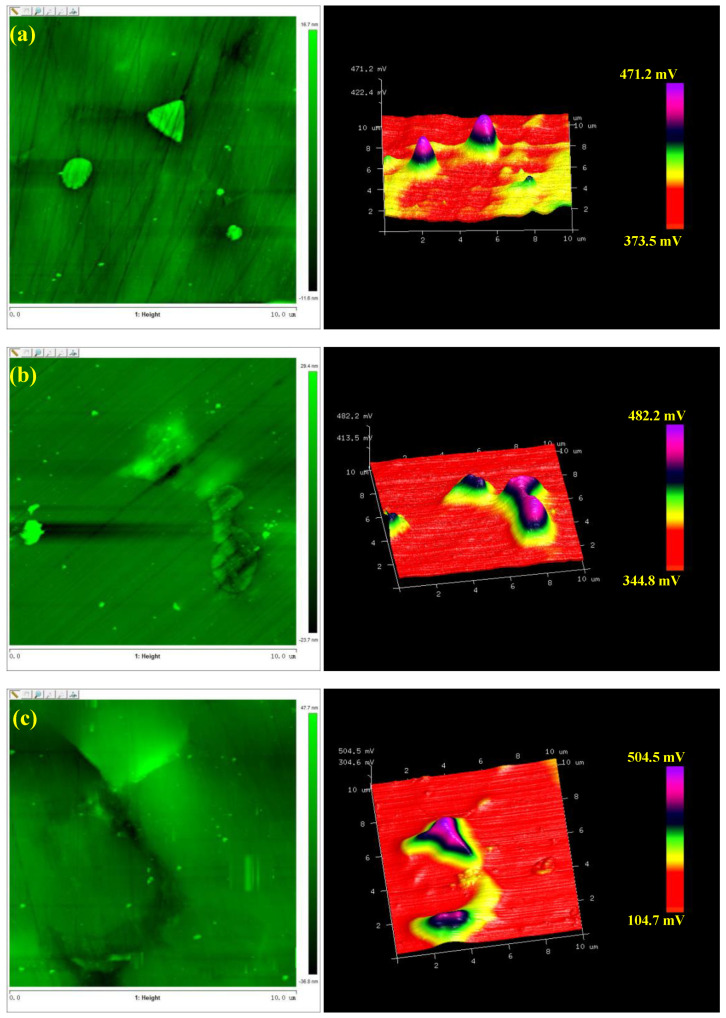
The morphology and potential diagram around Nd-rich phase of NdFeB magnet, (**a**) 5% Ce, (**b**) 3% Ce, (**c**) without Ce.

**Table 1 materials-13-04360-t001:** Chemical composition of the (Nd, Ce)FeBmagnets specimens (wt%).

Element	Nd	Fe	Pr	Zr	B	Ce	B
(a)	21.21	69.22	6.88	0.07	1.21	5.02	Bal
(b)	22.84	69.82	7.01	0.07	1.32	3.11	Bal
(c)	23.83	68.22	7.22	0.08	1.31	0	Bal

**Table 2 materials-13-04360-t002:** Parameters extracted from the polarization data of the (Nd,Ce)FeB magnets in 3.5% NaCl solution.

Ce Content/wt%	E_corr_/mV	I_corr_/Acm^−2^	β_a_/mV	β_c_/mV
(a) 5.0	−603.1	1.9 × 10^−^^8^	75.3	−136.8
(b) 3.0	−687.2	2.3 × 10^−^^8^	73.1	−138.7
(c) 0.0	−756.3	3.6 × 10^−^^9^	73.2	−138.8

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
