# Peer review of "Rare Earth Cerium Increases the Corrosion Resistance of NdFeB Magnets"

_materials, 2020, doi:10.3390/ma13194360_

Round 1

Reviewer 1 Report

The idea of using Ce (which is very abundant compared to the other rare earth elements) to replace Nd in magnets to increase corrosion resistance is intriguing and timely.

I was very curious about the effect on the magnetism.  While the authors demonstrate that the use of Ce to replace Nd in these magnets results in corrosion resistance in 3.5% NaCl solution, I did not see any mention of whether that same use of Ce also resulted in any loss of magnetism.  I imagine that many reading this would get excited about the increase in corrosion resistance, but also wonder about the effect on the magnetism of the magnetic materials.  The study in itself seems complete in its investigation of the corrosion resistance, and certainly the question of any change in magnetism would be appropriate for another study, but the authors might mention that the effect on the magnetism of the materials was investigated separately, or was a future study, or something.  After all, from a practical standpoint, increasing corrosion resistance is wonderful, especially using a cheaper or more abundant element, but if that results in significant loss of magnetism, then that's another problem...

There were some places where the English was hard to follow.  For the most part, I could follow what the authors were trying to get across, but in several places, I couldn't quite understand what they were trying to communicate:

  1. Line 84, indicating "the powder was ground using air under the protection of N2".  If air was present, what is the point of the N2?  I may be missing something, but usually the point of using N2 is to protect from air and/or water...but if air is specifically being used to contact the material, is the use of N2 accomplishing anything?
  2. Lines 203-204.  I think perhaps two separate sentences each got partially cut, and then accidentally spliced together.  The sentence as written doesn't make sense.
  3. In Figure 5, the label "Falling down of Nd2Fe14B phase" is used.  In English, that phrase doesn't seem to apply to the situation.  Perhaps another choice of phrase would make this more clear.

There were a number of typos present:

  1. Line 23, "equipmentin"--->"equipment in"
  2. Line 42, "with in"--->"within"
  3. Line 46, the elements copper, zinc, and vanadium all have uppercase letters in their names, when they should be lowercase like all the other elements listed.
  4. Line 47, "wolfram"--->"tungsten" (in English, although the IUPAC symbol is W, the IUPAC name is tungsten).
  5. Line 47, "zriconium"--->"zirconium"
  6. Line 64, "Yuet al."--->"Yu et al."
  7. Line 72, "naturally formed".  All the rare earth elements are naturally formed.  They are all also naturally occurring (with the possible exception of Pm, the isotopes of which are all unstable).  It might read more clearly to leave out "naturally formed"--the authors' point is still correct without the words "naturally formed".
  8. Line 175, the chemical reaction shown needs arrows, not equal signs. (Perhaps that was a problem with the electronic download and formatting?)
  9. Line 226, "Where;"--->"Where:"

Author Response

We sincerely thank the reviewers for the thoughtful and constructive comments and questions, and we also appreciate the editor for the work on processing the manuscript. All comments and questions have been discussed and replied in the following one by one. The changes, deletions and additions in the revised manuscript were marked by red color fonts.

Sincerely yours,

Zixuan Yang

Reviewer 2 Report

The paper is well written. Some minor editing correction should be done (e.g. spaces between values and units), but apart from that I think that it can be published in a present form.

Author Response

(The authors gave the same response as above.)

Reviewer 3 Report

Rare earth Cerium increase the corrosion resistance

The authors studied the effects of Ce on the corrosion resistance of NdFeB magnets. The authors found out that Ce-addition decreased the potential difference between the main phase and (Nd, Ce)-rich phase, which resulted in improved corrosion resistance.

The novelty of the article is questionable. A similar article is “Effect of cerium on the corrosion behavior of sintered (Nd,Ce)FeB magnet”; https://doi.org/10.1016/j.jmmm.2017.01.094. The authors are advised to explain in detail what the main differences between the articles are, and what is new in their article.

Otherwise, the article needs a substantial revision both in scientific writing and also in English. Some parts are not written sufficiently well.

There are also other topics that should be improved and are given below.

  • Line 2: Title: not »increase« but “increases”

Introduction

  • Line 23: "equipmentin" ?
  • Line 29-30: What is the composition of the Nd-rich phase?
  • Line 39: What is a quadrilateral phase?
  • Line 46: some elements with the large first letter other with the small
  • Lines 47 and 48: What means »m«?
  • Line 78: “EDS mapping” probably better “EDS-analysis”

Experiment

  • Line 84: »then the powder was ground using air under the protection of N2«. What means »using air«?
  • Line 85: The meaning of this sentence is not clear: »After the magnetic field orientation pressing shape.«
  • Line 91: How were the chemical compositions of alloys in Table 1 determined? Why the authors added 5 and 3 wt. % of Ce?
  • Line 94: Please use spaces between numbers and units (e.g. 40 mm).
  • ine 119: »were detected« replace with e.g. »were investigated.
  • Line 131: In Figure 1 replace a, b, and c with the Ce contents
  • Line 146: »PotentioCenamic«?
  • Line 151: »in Fig. 5.« Probably »in Fig. 3«
  • Line 157: »The presence of the inductive loop showed the manifestation of adsorption and desorption for the transition action in electrochemical corrosion«. Where can be seen inductive loops?
  • Line 199: Nb was not given in the initial chemical composition. EDS point analysis of phases could give a better indication of the chemical composition of different phases.
  • Line 205: »When the Ce content was increased to 5%, the grain boundary became relatively small and evenly distributed«. This sentence seems to be confused. A grain boundary cannot be small and evenly distributed!
  • Equations: The equations were probably obtained from the literature, but no references were given.
  • The results of metallography were not presented in sufficient details.

Author Response

(The authors gave the same response as above.)

Reviewer 4 Report

This work entitles “Rare earth Cerium increase the corrosion resistance of NdFeB magnets” by Dai et al. reported the mechanism of less corrosive NdFeB magnets by doping more readily available rare earth metal, which is Ce. Here in, the author demonstrates significant decrease of the corrosion tendency of NdFeB as the amount of Ce increases. The author has thoroughly analyzed and reviewed by conducting potentiodynamic polarization curves and Nyquist diagrams, where the larger diameter of the semicircle can refer to larger corrosion resistance, to prove the linear increase of the corrosion resistance depending on Ce content. Furthermore, EDS mapping and calculations are shown to support the proposed mechanism. The claims proposed by this work are certainly interesting, yet there is some missing information that the reviewer suggest addressing following queries before publication. 

Comment 1:

In the introduction, the author mentioned, “the B-rich and Nd-rich phases are generally distributed on the grain boundary. As such, oxidation of the grain boundary results in its corrosion.” The characters of grain boundaries in oxide layers formed on substrates influence adhesion and friction behaviour, surface fracture and wear during high temperature steel processing. However, the effect of grain characters on the oxidation behaviour is not fully discussed here. It is suggested to add more explanation for a better understanding.

Comment 2:

In figure 1, the potential declined abruptly as the current density increases, why does this kind of tendency is shown in the graph. Furthermore, it is suggested to illustrate the behind properties of Ce for endowing this result.

Comment 3:

The low value for the corrosion current density indicates small corrosion rate. In figure 2, it is not clear how the experimental data support the following statement, “at more positive potential than the potential of the corrosion Ecorr, the addition of metal Ce leads the anode polarization curve to a smaller current density.” The current density of magnets without Ce appear to be smaller which can refer to smaller corrosion rate. It is suggested to address the

Comment 4:

The author conducted various measurements to clarify the effects of Ce content on the corrosion behavior i.e., electrochemical tests, SEM, EDS mapping and SKPFM. However to make sure the results, it would be better to show photographed image that corrosion is conducted under harsh conditions to prove that the corrosion rate is inhibited with Ce content.

Comment 5:

 At the first part of the paper, the author claimed that by replacing the rare-earth Nd with Ce, it is possible to bring huge economic benefit. How much percent of the rare-earths Nd included NdFeB magnets could be replaced by cerium? In this paper the maximum content of cerium is 5 %. If the NdFeB magnets contain more content of Ce, is there any side effect of other problems?

Comment 6:

As mentioned in this paper, the most oxidative Nd, B- phase is located on the boundary of grain. In this regard, the corrosion resistance can be improved by modifying the grain boundary phase composition and distribution. However, the EDS mapping image does not reveal that cerium is concentrated on the grain boundary, but it seems well distributed over every phase. Furthermore, it is hard to find the difference between magnets with 3% Ce and 5% Ce. Thus, it is suggested to clarify the concentration and distribution of Ce on the grain boundary as a data attached in the ref. Journal of Alloys and Compounds 2017, 692, 190-197 (Fig. 3).

Author Response

(The authors gave the same response as above.)

Round 2

Reviewer 3 Report

The authors have made only minor modifications of the manuscript.

They did not explain what was the novelty of their article in comparison to

"Effect of cerium on the corrosion behavior of sintered (Nd,Ce)FeB magnet”; https://doi.org/10.1016/j.jmmm.2017.01.094.

Therefore, I do not not recommend this article for publishing in Metals in the current form.

Author Response

(The authors gave the same response as above.)
